# Serum Complement C4 Levels Are a Useful Biomarker for Predicting End-Stage Renal Disease in Microscopic Polyangiitis

**DOI:** 10.3390/ijms241914436

**Published:** 2023-09-22

**Authors:** Shogo Matsuda, Katsumasa Oe, Takuya Kotani, Ayana Okazaki, Takao Kiboshi, Takayasu Suzuka, Yumiko Wada, Hideyuki Shiba, Kenichiro Hata, Takeshi Shoda, Tohru Takeuchi

**Affiliations:** Department of Internal Medicine IV, Division of Rheumatology, Osaka Medical and Pharmaceutical University, Osaka 569-8686, Japan; shogo.matsuda@ompu.ac.jp (S.M.); takayasu.suzuka@ompu.ac.jp (T.S.); tooru.takeuchi@ompu.ac.jp (T.T.)

**Keywords:** anti-neutrophil cytoplasmic antibody-associated vasculitis, microscopic polyangiitis, end-stage renal disease

## Abstract

This study aimed to evaluate the risk factors for end-stage renal disease (ESRD) in microscopic polyangiitis (MPA). In total, 74 patients with MPA were enrolled, and we compared the baseline clinical characteristics and disease activity between MPA patients who have progressed to ESRD and those without ESRD to select predictive factors for ESRD. Out of 74 patients, 12 patients (16.2%) had ESRD during follow-up. Serum C4 levels were significantly higher in MPA patients who have progressed to ESRD than in those without ESRD (*p* = 0.009). Multivariate analyses revealed that high serum creatinine levels (odds ratio (OR) 4.4, 95% confidence interval (CI) 1.25–15.5) and high serum C4 levels (OR 1.24, 95% CI 1.03–1.49) were risk factors for ESRD. Using receiver operating characteristic analysis, the cut-off value for initial serum C4 levels and serum creatinine levels were 29.6 mg/dL and 3.54 mg/dL, respectively. Patients with MPA with a greater number of risk factors (serum C4 levels > 29.6 mg/dL and serum creatinine levels > 3.54 mg/dL) had a higher ESRD progression rate. Serum C4 levels were significantly positively correlated with serum creatinine levels and kidney Birmingham vasculitis activity score (*p* = 0.02 and 0.04, respectively). These results suggest that serum C4 levels are useful tools for assessing renal disease activity and prognosis in MPA.

## 1. Introduction

Anti-neutrophil cytoplasmic antibody (ANCA)-associated vasculitis (AAV) is defined by the International Chapel Hill Consensus Conference (CHCC) as necrotizing vasculitis with few or no immune deposits that predominantly affects small vessels [1].

AAV includes microscopic polyangiitis (MPA), granulomatosis with polyangiitis (GPA), and eosinophilic granulomatosis with polyangiitis. The major target antigens of ANCA are proteinase 3 (PR3) and myeloperoxidase (MPO) in AAV. MPA is commonly associated with MPO-ANCA, and GPA is mainly associated with PR3-ANCA [2]. MPA is a type of AAV that primarily affects the lungs and kidneys [3,4]. It has been reported that kidney involvement is the most common manifestation that occurs in 78.8% of MPA patients [5].

Renal involvement is a major prognostic factor for AAV [6,7,8]. Twenty-five percent of AAV patients develop end-stage renal disease (ESRD), even if they are treated with prednisolone and immunosuppressive therapy [9,10]. Several reports have shown that the risk factors for ESRD in AAV are renal histological findings of sclerotic glomeruli, thrombotic microangiopathies, and IgG depletion [11,12,13]. MPO-ANCA positivity is also a risk factor for ESRD in addition to renal histopathology [14]. In AAV with renal disease, MPO-ANCA-positive patients have worse renal survival than PR3-ANCA-positive patients [15].

A Japanese multicenter cohort study also showed that ESRD-free survival was much worse in patients with MPA than in those with GPA [16]. There were scoring systems to predict ESRD in AAV [17,18,19], but prognostic factors for ESRD in MPA have not been fully elucidated.

Activation of the complement system is associated with the pathogenesis of AAV [20]. In particular, activation of the alternative complement pathway (aCP) in the complement system is majorly associated with the pathogenesis of AAV with glomerulonephritis (GN) [21]. Complement C3 deposition in the kidney is associated with worse renal function in AAV-GN [22]. Inhibition of complement component C5 prevented the development of AAV-GN in mice [23], and avacopan (a C5a receptor inhibitor) could be a useful treatment for renal inflammation in AAV [24]. However, it is unclear whether activations of other complement pathways, such as the classical and mannose-binding lectin (MBL) pathways, are related to the pathomechanism of MPA-GN.

This study evaluated factors indicating poor prognosis and progression of ESRD in patients with MPA. We reported that high serum complements C4 and creatinine levels are useful biomarkers for assessing renal disease activity and prognosis in MPA.

## 2. Results

### 2.1. Patient Profiles

In total, 107 patients were classified as having MPA in this study. Of these, 33 patients were excluded because they did not have serum complements measured before remission induction therapy or had already reached ESRD on admission (Figure 1). The baseline characteristics of 74 patients with MPA are shown in Table 1. The median patient age was 75 (70–79) years, and 37 (50%) patients were women. MPO-ANCA was positive in 73 patients, whereas PR3-ANCA was positive in three patients. Two patients were double-positive for MPO/PR3-ANCA. The median serum creatinine levels were 1.3 (0.7–2.2) mg/dL, and the median serum C3 and C4 levels were 112 (93.8–132.3) mg/dL and 24.9 (18.4–31.4) mg/dL, respectively. The median total BVAS at onset was 18 (12–22), and the proportions of 2009 FFS ≤ 1, =2, and ≥3 were 20.3%, 47.3%, and 32.4%, respectively. The proportions of “localized”, “early systemic”, “systemic”, and “severe” as defined by the EUVAS classifications were 2.7%, 21.6%, 56.8%, and 18.9%, respectively. Seventy-three patients were treated with an initial remission induction therapy on admission. One patient experienced recurrence and had been treated with low-dose prednisolone (5 mg/day). Details regarding systemic organ involvement and treatments are shown in Appendix A.

### 2.2. Comparison of the Baseline Clinical Characteristics and Disease Severity Classification of MPA Patients Who Have Progressed to ESRD and Those without ESRD

Twelve patients (16.2%) developed ESRD during follow-up. The ESRD incidence rate was 5.9 per 100 patient-years, and the cumulative renal survival rate was 77% at 5 years. We compared the baseline clinical characteristics of patients with MPA who have progressed to ESRD and those without ESRD (Table 2). The initial WBC count and Hb level were significantly higher in MPA without ESRD than in MPA patients who have progressed to ESRD (*p =* 0.02 and *p* = 0.004, respectively). In addition, the initial serum creatinine levels were significantly higher in MPA patients who have progressed to ESRD (5.8 [3.7–9.3]) than in MPA patients without ESRD (0.95 [0.69–1.68]) (*p* < 0.0001). Baseline serum C3 levels were significantly lower in MPA patients who have progressed to ESRD (96.5 [79.3–115.8]) than in MPA patients without ESRD (114 [97.3–133]) (*p* = 0.03). In contrast, the baseline serum C4 levels were significantly higher in MPA patients who have progressed to ESRD (31.3 [24.1–40.6]) than in MPA patients without ESRD (24.2 [17.3–30.0]) (*p* = 0.009). The percentage of patients with 2009 FFS ≥ 3 and “severe” MPA as defined by the EUVAS classification was significantly higher in patients with MPA patients who have progressed to ESRD than in those without ESRD (*p* = 0.001, *p* < 0.0001, respectively). A comparison of the initial treatments of MPA patients who have progressed to ESRD and those without ESRD is shown in Table 3. Although no differences were observed in the doses of prednisolone, or the doses of total intravenous cyclophosphamide, or the frequency of administration of rituximab and immunosuppressant therapy between the two groups, the frequency of administration of methylprednisolone pulse therapy was higher in MPA patients who have progressed to ESRD compared to those without ESRD (*p* = 0.013).

### 2.3. Selection of Risk Factor for ESRD in MPA

Univariate analysis revealed that there was a significant difference in initial WBC count and serum levels of Hb, creatinine, C3, and C4 between the ESRD and non-ESRD groups. Next, we performed a multivariate analysis to identify the risk factors for ESRD. The multivariate analysis using these variables identified serum creatinine (OR 4.4, 95% CI 1.25–15.5) and C4 levels (OR 1.24, 95% CI 1.03–1.49) were independently associated with the risks for ESRD (*p* = 0.02, *p* = 0.02, respectively) (Table 4).

### 2.4. Renal Survival Rates in MPA

We performed ROC analysis to determine a cut-off value for initial serum C4 and creatinine levels, which indicate ESRD risk. The cut-off values for initial serum C4 and creatinine levels were set to 29.6 mg/dL (sensitivity: 0.75, specificity: 0.74, AUC: 0.74) and 3.54 mg/dL (sensitivity: 0.83, specificity: 0.95, AUC: 0.95), respectively. The patients were then divided into three groups based on the number of risk factors (serum C4 levels ≥ 29.6 mg/dL and serum creatinine levels ≥ 3.54 mg/dL), and Kaplan–Meier survival curves were plotted (Figure 2). The 5-year ESRD-free rate was 96.2% in the patients without risk factors, 60.5% in the patients with one risk factor, and 0% in those with two risk factors. The ESRD rate increased as the number of risk factors increased (*p* < 0.0001).

### 2.5. Comparison of the Baseline Clinical Characteristics and Disease Severity Classification of Patients with MPA with Serum C4 Levels above and below the Cut-Off Value

We compared the baseline clinical characteristics of MPA patients with serum C4 levels ≥ 29.6 mg/dL and <29.6 mg/dL (Table 5). The initial WBC count was significantly lower in MPA with serum C4 levels ≥ 29.6 mg/dL than in MPA with serum C4 levels < 29.6 mg/dL (*p =* 0.002). In addition, the initial albumin and serum creatinine levels were significantly higher in MPA with serum C4 levels ≥ 29.6 mg/dL than in MPA with serum C4 level < 29.6 mg/dL (*p* = 0.005 and *p* = 0.008, respectively).

### 2.6. Correlation between Serum C4 and Indicators of AAV Disease Activity

Next, we evaluated the correlation between serum C4 levels and renal disease activity in MPA patients (Figure 3). Serum C4 levels were significantly positively correlated with serum creatinine levels (R = 0.26, *p* = 0.02) (Figure 3A). Additionally, we compared the correlation between serum C4 levels and BVAS scores. Serum C4 levels were significantly positively correlated with kidney BVAS scores (R = 0.24, *p* = 0.04) (Figure 3B).

### 2.7. Comparison of Serum CRP Levels and C4 Levels between MPA Patients Who Have Progressed to ESRD and Those without ESRD and Control Group

To evaluate whether elevation of serum C4 levels is associated with systemic inflammation, we compared serum C4 levels and CRP levels between MPA patients who have progressed to ESRD and those without ESRD and the control group. Interestingly, serum CRP levels were significantly higher in MPA patients who have progressed to ESRD and those without ESRD compared to the control group (*p* < 0.0001), but there was no significance in serum CRP levels between MPA patients who have progressed to ESRD and those without ESRD (*p* = 0.22). (Figure 4A) On the other hand, serum C4 levels were significantly higher in MPA patients who have progressed to ESRD compared to those without ESRD and the control group (*p* = 0.026 and 0.001, respectively). There was no significance in serum C4 levels between MPA patients without the ESRD group and the control group (*p* = 0.37). (Figure 4B) This finding suggests that elevation of serum C4 levels cannot be explained only by systemic inflammation in MPA.

## 3. Discussion

This study revealed that MPA patients who have progressed to ESRD had lower WBC, Hb, and serum C3 levels and higher serum creatinine and C4 levels than did MPA patients without ESRD. In multivariate analysis, serum C4 levels and serum creatinine levels were independently extracted as risk factors for ESRD in MPA patients. Additionally, high serum C4 levels were positively correlated with renal disease severity in patients with MPA. Thus, serum C4 levels were a predictive biomarker for evaluating renal disease severity in MPA.

In this study, 16.2% of MPA developed ESRD during follow-up, and the cumulative renal survival rate was 77% at 5 years. The previous multicenter cohort studies with MPA patients and MPA-dominant-AAV showed cumulative renal survival rates of 70% to 80% at 5 years [25,26,27]. These data are consistent with our results.

Previous reports have shown that initial serum creatinine levels, baseline eGFR decrease, and lower serum C3 levels are risk factors for ESRD and death in AAV [9,28,29,30]. Serum C3 hypocomplementemia was caused by C3 consumption in the kidney due to the activation of aCP pathway in AAV-GN [31]. This study also showed that serum creatinine levels are significantly higher and serum C3 levels were lower in the ESRD group compared to those in the non-ESRD group, which supports the results of previous studies.

Complement C4 plays an important role in the activation of the classical and MBL pathways [32]. Several reports have shown that serum C4 levels cannot predict renal survival in patients with AAV, including MPA and GPA patients [33,34,35]. However, few studies focus on the association of serum C4 levels with ESRD in MPA-GN.

In this study, we first revealed that high serum C4 levels are associated with the progression of ESRD and correlate with renal disease severity in MPA-GN. Gou et al. previously reported that urinary levels of C1q in patients with active MPO-ANCA positive AAV were significantly higher than those in the control groups, suggesting the activation of the classical pathways in AAV [36]. Additionally, Kojima et al. previously reported that the classical pathway is associated with the pathomechanism of MPO-ANCA-associated glomerulonephritis (AAGN) [37]. They showed that circulating immune complexes (CICs) were found in 65% of MPO-AAGN, and CICs activate the classical complement pathway, leading to the formation of C5b-9 in MPO-AAGN [37]. Based on these findings, activation of the classical pathway, which increases serum C4 levels [38], may be associated with the pathomechanism in MPA-GN. In other glomerular nephropathies, including IgA nephropathy and diabetic nephropathy, high serum C4 levels are associated with renal disease severity. Pan et al. reported that there was a significantly negative correlation between serum C4 levels and eGFR at baseline in IgA nephropathy [39]. Additionally, a positive correlation between serum C4 levels and urinary protein levels was reported in diabetic nephropathy [40]. These reports were consistent with our findings, showing that there were positive correlations between serum C4 levels, serum Cr levels, and kidney BVAS levels. Serum C4 levels are a surrogate marker to evaluate the severity of renal damage in patients with MPA-GN.

In our study, baseline serum C3 levels were significantly lower, but the baseline serum C4 levels were significantly higher in MPA patients who have progressed to ESRD than in MPA patients without ESRD. One possible reason is that there are production and consumption balance differences between C3 and C4. Lower C3 levels were associated with poorer renal survival because activation of alternative pathways leads to the deposition of C3 in AAV-GN, leading to the consumption of C3 [22,41,42]. In contrast, smaller amounts of C4 deposition were detected compared to those of C3 deposition in the kidney of MPO/PR3-ANCA-positive patients with crescentic glomerulonephritis [43]. In addition, classical and MBL pathways activation causes kidney injury, leading to the production of C4 from renal macrophage, glomerular epithelium, and renal tubular epithelium [44,45,46]. These results suggest that C4 production is activated and C4 consumption is less in the kidney, showing that high C4 levels may be associated with poor renal prognosis in AAV.

In our study, we divided MPA patients into three subgroups based on the number of risk factors, including serum C4 levels and serum creatinine levels, and we found that the ESRD rate increased as the number of risk factors increased. Previous reports showed that serum creatinine level is a useful marker for predicting ESRD in MPA [47,48]. On the contrary, it has not been elucidated that serum C4 levels can predict the development of ESRD in patients with MPA. However, several reports showed that stratification of patients using serum C4 levels was useful for predicting ESRD in other renal diseases [38,40,49]. First, Liu et al. reported that patients with idiopathic membranous nephropathy with high C4 levels had a significantly lower cumulative incidence of renal survival compared to those with low and median C4 levels [38]. Second, Duan et al. reported that patients with diabetic nephropathy with higher serum C4 levels had significantly faster deterioration of renal function compared to those with lower serum C4 levels [40]. Third, Bi et al. showed that patients with IgA nephropathy with high serum C4 levels had poorer prognosis, including ESRD and a ≥50% reduction in eGFR, compared to those with low and normal serum C4 levels [49]. These results support our findings, and the classification of MPA patients using serum Cr levels and C4 levels at baseline may be useful for predicting ESRD in MPA patients. However, further studies are needed to elucidate the pathogenesis of the association of high C4 levels with the risk of developing renal injury in MPA-GN.

Several biomarkers, including serum neutrophil gelatinase-associated lipocalin, microRNAs, and urinary MCP-1, have been reported to correlate with renal disease activity in patients with AAV [50,51,52]. However, there have been few reports showing the relationship between biomarkers and ESRD in AAV [53,54]. Our study showed that the optimal cutoff of baseline serum C4 levels for ESRD progression was set at 29.6 mg/dL, and measuring whether baseline serum C4 level is over this cutoff level may be an invasive and useful method for predicting ESRD in MPA. However, we have not clarified whether serum C4 level is a more sensitive predictor of ESRD compared to other biomarkers in this study. Therefore, further studies are needed to prove this.

Our study had some limitations. First, this was a retrospective study, and the sample size was limited. Second, all patients in our study were Japanese and MPA; therefore, it remains unknown whether this finding is applicable to other ethnicities or other types of AAV. Third, we could not evaluate complement deposition in the renal pathology. Fourth, our study showed a statistically significant correlation between serum C4 levels and serum creatinine levels, as well as between serum C4 levels and kidney BVAS scores, but there was a weak correlation between them. Additionally, serum creatinine levels are major components in kidney BVAS scores, so the correlation between serum C4 levels and kidney BVAS scores may be influenced by serum creatinine levels [55]. Further study is needed to understand whether high serum C4 levels reflect renal disease severity in patients with MPA. Finally, high serum C4 levels were a more predictive biomarker for ESRD than low C3 levels in MPA, but further investigation using multi-center studies is needed to determine which of the two is a more predictive biomarker in MPA-GN.

## 4. Materials and Methods

### 4.1. Patients

Patients admitted to the Osaka Medical and Pharmaceutical University between December 2010 and June 2021 were enrolled in this study. All patients were diagnosed with AAV according to the CHCC definition [1]. Patients received immunosuppressive treatment at the attending physician’s discretion. The exclusion criteria were as follows: (1) malignancy, (2) infection, (3) drug-induced vasculitis, (4) secondary vasculitis, (5) vasculitis mimics, and/or (6) sarcoidosis [56]. Clinical data were obtained from the patients’ medical records upon admission. For the control group, 32 subjects were recruited following denial of complement-related inflammatory autoimmune diseases, such as systemic lupus erythematosus and cryoglobulinemic vasculitis.

### 4.2. Measurement of Clinical Signs and Laboratory Parameters

We evaluated the patients’ demographic characteristics (age and sex) and the treatments used. Measurements of white blood cell (WBC) count, hemoglobin (Hb), albumin, creatinine, CRP, MPO-ANCA, and PR3-ANCA were obtained. Serum MPO-ANCA and PR3-ANCA titers were measured using an enzyme-linked immunosorbent assay. Serum complement C3 and C4 levels were measured using turbidimetric immunoassay.

### 4.3. Evaluation of Systemic Disease Severity

Disease severity was determined according to the European Vasculitis Study Group (EUVAS) classifications [57], and organ involvement was evaluated using the Birmingham vasculitis activity score (BVAS) version 3.0 [55]. The 2009 Five-Factor Score (FFS), which is used to evaluate prognosis at MPA diagnosis, was completed for each patient [58].

### 4.4. Renal Outcome

End-stage renal disease (ESRD) was defined as an estimated glomerular filtration rate (eGFR) < 15 mL/min/1.73 m^2^ and a requirement for permanent renal replacement therapy. Patients who were dependent on hemodialysis from the time of MPA diagnosis for more than 3 months were considered as having ESRD, as previously described [41]. We assessed ESRD development until death or the last follow-up date (31 December 2021).

### 4.5. Statistical Analysis

Data are presented as the median and interquartile range. Fisher’s exact test was used when appropriate, and the Mann–Whitney U test was used to compare median values. Steel–Dwass test was used for multiple comparisons. To identify poor prognostic factors for ESRD, a multivariate logistic regression model was conducted, including all variables that were significantly different in univariate analyses. We calculated Odds ratios (OR) and 95% confidence intervals (CI) to examine independent risk factors for ESRD. We then performed a receiver operating characteristic (ROC) curve analysis to determine the most suitable cut-off level for these risk factors. The Kaplan–Meier method was used to assess survival curves, and the log-rank test was used to evaluate the significance of differences between the three groups. Correlations were evaluated using Spearman’s rank correlation coefficient. Statistical significance was set at *p* < 0.05. Data were analyzed using JMP statistical software (version 15.0; SAS Institute Inc., Cary, NC, USA) and GraphPad Prism (version 8.0; GraphPad Software, La Jolla, CA, USA).

## 5. Conclusions

We revealed that high serum C4 levels are predictive biomarkers for ESRD in MPA patients. In addition, we revealed that high serum C4 levels are associated with renal disease severity, as evaluated by serum creatinine levels and kidney BVAS. Prediction of ESRD stratifies by serum C4 will help rheumatologists manage MPA-GN. Further investigations are needed to elucidate the pathomechanism of serum C4 activation in MPA-GN.

## Figures and Tables

**Figure 1 ijms-24-14436-f001:**
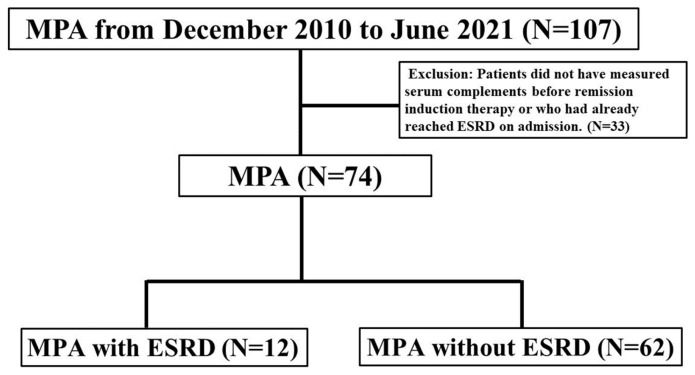
Flowchart of this study. Flowchart illustrating the selection and the prognosis of patients with MPA.

**Figure 2 ijms-24-14436-f002:**
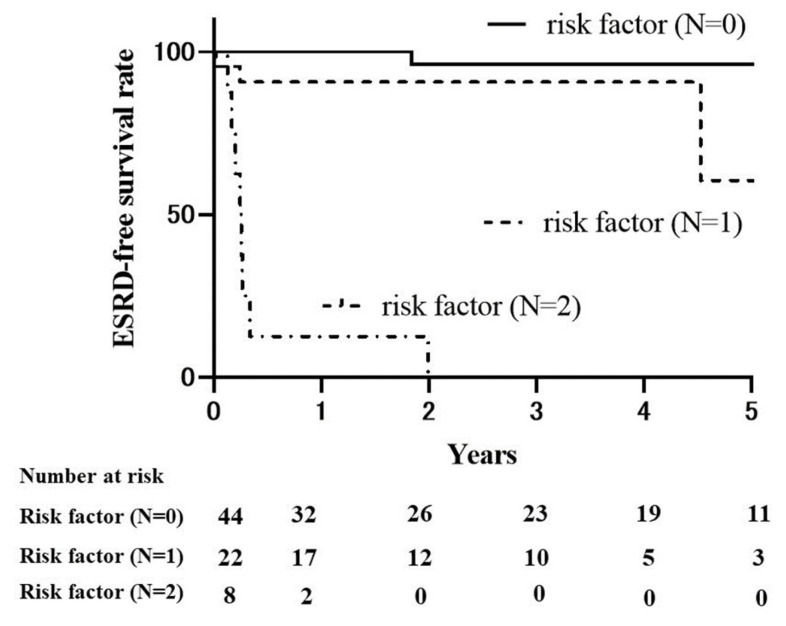
ESRD-free survival curves for MPA patients based on initial serum C4 and creatinine levels. The MPA patients were divided into three groups based on the number of risk factors (serum C4 levels ≥ 29.6 mg/dL and serum creatinine levels ≥ 3.54 mg/dL). The 5-year ESRD-free rate was 96.2% in the patients without risk factors, 60.5% in the patients with one risk factor, and 0% in those with two risk factors. The risk of ESRD increases with the number of risk factors (*p* < 0.0001). Solid line: group without risk factors, dashed line: group with 1 risk factor, dot-dash line: group with 2 risk factors. ESRD: end-stage renal disease; MPA: microscopic polyangiitis.

**Figure 3 ijms-24-14436-f003:**
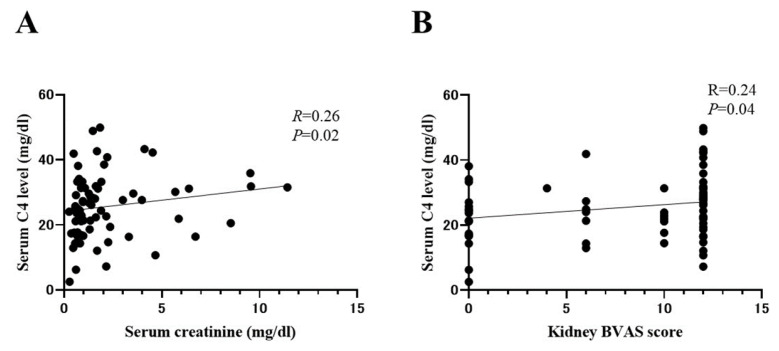
Relationship between serum C4 levels and kidney disease activity indicators in MPA. (**A**) Correlation between serum C4 levels and serum creatinine levels. (**B**) Correlation between serum C4 levels and kidney BVAS scores. Correlations were evaluated using Spearman’s rank correlation coefficient. Statistical significance was set at *p* < 0.05.; BVAS: Birmingham vasculitis activity score.

**Figure 4 ijms-24-14436-f004:**
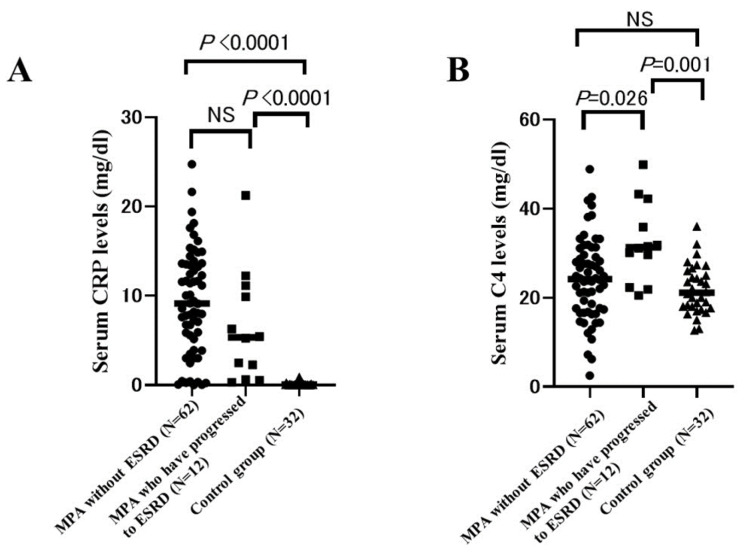
Comparison of serum CRP levels and C4 levels between MPA patients who have progressed to ESRD and those without ESRD and the control group. (**A**) Comparison of serum CRP levels between MPA patients who have progressed to ESRD and those without ESRD and the control group. (**B**) Comparison of serum C4 levels between MPA patients who have progressed to ESRD and those without ESRD and the control group. Steel–Dwass test was used for multiple comparisons. A *p*-value of <0.05 was considered significant. The horizontal bar represents the mean value.

**Table 1 ijms-24-14436-t001:** Clinical characteristics, and disease severity classification of patients in MPA.

Characteristics	AAV (N= 74)
Age, years	75 (70–79)
Female, n (%)	37 (50)
Laboratory findings	
WBC,/mm^3^	11,315 (7980–14,705)
Hb, g/dL	9.8 (8.7–11.8)
Alb, g/dL	2.5 (2.1–3.1)
Cr, mg/dL	1.3 (0.7–2.2)
CRP, mg/dL	8.4 (3.8–13.3)
Serum C3 levels, mg/dL	112 (93.8–132.3)
Serum C4 levels, mg/dL	24.9 (18.4–31.4)
Positive, anti-MPO-ANCA, n (%)	73 (98.6)
Positive, anti-PR3-ANCA, n (%)	3 (4.1)
MPO-ANCA titer, U/mL	128.6 (65.0–280.0) ^a^
BVAS at onset	18 (12–22)
Five factor score 2009	
≤1	15 (20.3)
2	35 (47.3)
≥3	24 (32.4)
EUVAS-defined disease activity	
Localized	2 (2.7)
Early systemic	16 (21.6)
Systemic	42 (56.8)
Severe	14 (18.9)

The laboratory markers are presented as the median (interquartile range). The *p*-values were estimated using Fisher’s exact test or the Wilcoxon rank sum test. ^a^ Number of subjects, n = 73. MPA: microscopic polyangiitis; AAV: anti-neutrophil antibody-associated vasculitis; WBC: white blood cell; Hb: hemoglobin; Alb: albumin; Cr: creatinine; CRP: C-reactive protein; MPO-ANCA: myeloperoxidase-anti-neutrophil cytoplasmic autoantibody; PR3-ANCA: proteinase 3-anti-neutrophil cytoplasmic antibody; BVAS: Birmingham Vasculitis Activity Score; EUVAS: European Vasculitis Study Group.

**Table 2 ijms-24-14436-t002:** Comparison of baseline clinical characteristics and disease severity classification of patients in MPA patients who have progressed to ESRD and those without ESRD.

Characteristics	MPA Who Have Progressed to ESRD (N = 12)	MPA without ESRD (N = 62)	*p*-Value
Age, years	72 (70–77)	76 (70–79)	0.52
Female, n (%)	4 (33.3)	33 (53.2)	0.34
Laboratory findings			
WBC,/mm^3^	8005 (5155–11,793)	11,870 (8698–15,510)	0.02 *
Hb, g/dL	9.1 (7.5–9.5)	10.4 (9.0–12)	0.004 **
Alb, g/dL	2.8 (2.5–3.4)	2.4 (2.1–3.0)	0.06
Cr, mg/dL	5.8 (3.7–9.3)	0.95 (0.69–1.68)	<0.0001 ***
CRP, mg/dL	5.3 (1.0–10.8)	9.1 (4.8–13.6)	0.095
Serum C3 level, mg/dL	96.5 (79.3–115.8)	114 (97.3–133)	0.03 *
Serum C4 level, mg/dL	31.3 (24.1–40.6)	24.2 (17.3–30.0)	0.009 **
Positive, anti-MPO-ANCA, n (%)	12 (100)	61 (98.4)	1.00
Positive, anti-PR3-ANCA, n (%)	0 (0)	3 (4.9)	1.00
MPO-ANCA titer, U/mL	94.4 (57.3–572.5)	143(68.7–280) ^a^	0.78
BVAS at onset	19 (18–22.8)	17 (11.8–21.0)	0.09
Five factor score 2009			
≤1	0 (0)	15 (24.2)	0.11
2	3 (25.0)	32 (51.6)	0.12
≥3	9 (75.0)	15 (24.2)	0.001 **
EUVAS-defined disease activity			
Localized	0 (0)	2 (3.2)	1.00
Early systemic	0 (0.0)	16 (25.8)	0.058
Systemic	4 (33.3)	38 (61.3)	0.11
Severe	8 (66.7)	6 (9.7)	<0.0001 ***

The laboratory markers are presented as the median (interquartile range). The *p*-values were estimated using Fisher’s exact test or the Wilcoxon rank sum test. ^a^ Number of subjects, n = 61. * *p* < 0.05, ** *p* < 0.01, *** *p* < 0.001. MPA: microscopic polyangiitis; ESRD: end-stage renal disease; WBC: white blood cell; Hb: hemoglobin; Alb: albumin; Cr: creatinine; CRP: C-reactive protein; MPO-ANCA: myeloperoxidase-anti-neutrophil cytoplasmic autoantibody; PR3-ANCA: proteinase 3-anti-neutrophil cytoplasmic antibody; BVAS: Birmingham Vasculitis Activity Score; EUVAS: European Vasculitis Study Group.

**Table 3 ijms-24-14436-t003:** Comparison of initial treatment of patients in MPA patients who have progressed to ESRD and those without ESRD.

Characteristics	ESRD (N= 12)	Non-ESRD (N= 62)	*p*-Value
Initial treatment			
PDN, mg/day	50 (40–60)	50 (36.9–60)	0.46
MPDN pulse, n (%)	7 (58.3)	13 (21.0)	0.013 *
Immunosuppressants			
IVCY, n (%)	8 (66.7)	33 (53.2)	0.53
Total IVCY dose (g)	1.2 (0.18–1.5)	1.4 (0.65–2.1)	0.09
RTX, n (%)	3 (25.0)	7 (11.3)	0.35
IVIG, n (%)	0 (0)	2 (3.2)	1.0
AZA/MTX/TAC/MZB, n (%)	6 (50.0)/0 (0)/0 (0)/0 (0)	44 (71.0)/2 (3.2)/4 (6.5)/1 (1.6)	0.19/1.0/1.0/1.0

The laboratory markers are presented as the median (interquartile range). * *p* < 0.05. MPA: microscopic polyangiitis; ESRD: end-stage renal disease; PDN: prednisolone; MPDN: methylprednisolone; IVCY: intravenous cyclophosphamide; RTX: rituximab; IVIG: intravenous immunoglobulin; AZA: azathioprine; MTX: methotrexate; TAC: tacrolimus; MZB: mizoribine.

**Table 4 ijms-24-14436-t004:** Risk factors of ESRD in patients with MPA.

Variable	Odds Ratio	Odds Ratio (95% CI)	*p*-Value
Serum C3 levels (for 1 mg/dL)	0.98	0.92–1.05	0.65
Serum C4 levels (for 1 mg/dL)	1.24	1.03–1.49	0.02 *
Serum Cr levels (for 1 mg/dL)	4.40	1.25–15.5	0.02 *
WBC (for 1/mm^3^)	0.99	0.99–1.00	0.37
Hb (for 1 g/dL)	0.35	0.12–1.01	0.053

A multivariate logistic regression model. A *p* value of <0.05 was considered significant. * *p* < 0.05. ESRD: end-stage renal disease; MPA: microscopic polyangiitis; Cr: creatinine; WBC: white blood cell; Hb: hemoglobin.

**Table 5 ijms-24-14436-t005:** Comparison of clinical characteristics and disease severity classification of patients in MPA with serum C4 levels ≥ 29.6 mg/dL and <29.6 mg/dL.

Characteristics	C4 Levels < 29.6 mg/dL(N = 49)	C4 Levels ≥ 29.6 mg/dL(N = 25)	*p*-Value
Age, years	73 (69–79)	76 (71–79)	0.55
Female, n (%)	27 (55.1)	10 (40.0)	0.33
Laboratory findings			
WBC,/mm^3^	12,270 (9005–15,845)	8800 (6155–12,145)	0.002 **
Hb, g/dL	9.7 (8.5–11.1)	10 (9.1–12.3)	0.16
Alb, g/dL	2.3 (2.0–2.9)	2.8 (2.3–3.7)	0.005 **
Cr, mg/dL	0.95 (0.66–1.79)	1.73 (0.94–4.32)	0.008 **
CRP, mg/dL	9.9 (6.3–13.5)	5.2 (0.5–11.9)	0.038 *
Serum C3 level, mg/dL	113 (89.5–131.5)	108 (96.5–132.5)	0.87
MPO-ANCA titer	120.2 (61.2–256.0) ^a^	223 (72.0–292.6)	0.20
BVAS at onset	18 (12–21.5)	17 (12–22)	0.60
Five factor score 2009			
≤1	13 (26.5)	2 (8.0)	0.07
2	22 (44.9)	13 (52.0)	0.63
≥3	14 (28.6)	10 (40.0)	0.43
EUVAS-defined disease activity			
Localized	0 (0)	2 (8.0)	0.11
Early systemic	11 (22.5)	5 (20.0)	1.00
Systemic	30 (61.2)	12 (48.0)	0.33
Severe	8 (16.3)	6 (24.0)	0.53

The laboratory markers are presented as the median (interquartile range). The *p*-values were estimated using Fisher’s exact test or the Wilcoxon rank sum test. ^a^ Number of subjects, n = 48. * *p* < 0.05, ** *p* < 0.01. MPA: microscopic polyangiitis; WBC: white blood cell; Hb: hemoglobin; Alb: albumin; Cr: creatinine; CRP: C-reactive protein; MPO-ANCA: myeloperoxidase-anti-neutrophil cytoplasmic autoantibody; BVAS: Birmingham Vasculitis Activity Score; EUVAS: European Vasculitis Study Group.

## Data Availability

The raw data supporting the conclusions of this article will be made available by the authors, without undue reservation.

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
