# Peer review of "Serum Complement C4 Levels Are a Useful Biomarker for Predicting End-Stage Renal Disease in Microscopic Polyangiitis"

_ijms, 2023, doi:10.3390/ijms241914436_

Round 1
Reviewer 1 Report
The authors present a retrospective study in which they demonstrate the usefulness of the C4 assay for predicting renal failure in microscopic polyangiitis-
Remarks:
1) The statement "patients with ESRD" is not clear as it seems that patients were studied at baseline when they already had end-stage renal disease. Better "patients who have progressed to ESRD" throughout the text
2) "avacopan" and "inhibitor" should not be put in italics
3) It should be clarified in Table 2 that the patients' values are relative to baseline and not to those in the ESRD phase, as it might appear.
4) In Table 3, creatinine values are a risk factor related to progression to ESRD. Discuss further in detail whether there is a correlation between elevated creatinine, and thus more severe renal damage, and higher C4 values already at baseline. This is discussed only in part in the section on study limitations. Clarify in Table 4 that the values reported refer to the baseline.
5) A table on the treatment received by patients who did or did not progress to ESRD and whether this differed statistically between the two groups should be added in the main text rather than in the supplementary data.
Author Response
Each response is written following the reviewers’ comments and all revisions are indicated in red font in the revised manuscript. Numbers of page and line correspond to revised version.
Comments of Reviewer #1
- The statement "patients with ESRD" is not clear as it seems that patients were studied at baseline when they already had end-stage renal disease. Better "patients who have progressed to ESRD" throughout the text
Response: Thank you very much for your important comments. Our previous writing style can be misleading to the readers, so we changed from “patients with ESRD” to “patients who have progressed to ESRD” in the main text and Figure 4.
- "avacopan" and "inhibitor" should not be put in italics
Response: Thank you very much for your comments. We changed these italic types to normal types. (Page 2 Line 60)
- It should be clarified in Table 2 that the patients' values are relative to baseline and not to those in the ESRD phase, as it might appear.
Response: Thank you very much for your helpful comments. As you pointed out, we measured the patients' values at baseline. Therefore, we changed this sentence as follows: “We compared the baseline clinical characteristics of patients with MPA, who have progressed to ESRD and those without ESRD (Table 2).” (Page 4 Line 98-99, 102-103) We also changed the title in Table 2.
- In Table 3, creatinine values are a risk factor related to progression to ESRD. Discuss further in detail whether there is a correlation between elevated creatinine, and thus more severe renal damage, and higher C4 values already at baseline. This is discussed only in part in the section on study limitations. Clarify in Table 4 that the values reported refer to the baseline.
Response: Thank you very much for your helpful comments. In other glomerular nephropathies, including IgA nephropathy and diabetic nephropathy, high serum C4 levels are associated with renal disease severity. Pan, et al. reported that there was a significantly negative correlation between serum C4 levels and eGFR at baseline. (Pan, M, et al. BMC Nephrol. 2017 Jul 11;18(1):231.) Also, the positive correlation between serum C4 levels and urinary protein levels were reported in diabetic nephropathy. (Duan, S, et al. Front Immunol. 2020 Sep 2;11:2073.) These reports were consistent with our findings, showing that there were positive correlations between serum C4 levels and serum Cr levels and Kidney BVAS levels. We added it on the discussion section. (Page 10 Line 257-263)
Additionally, as you pointed out, we compared the patients' values at baseline between patients with MPA with serum C4 levels above and below the cut-off value. We added “baseline” in result section. (Page 7 Line 171,173)
5) A table on the treatment received by patients who did or did not progress to ESRD and whether this differed statistically between the two groups should be added in the main text rather than in the supplementary data.
Response: According to the reviewer’s advice, we added Supplementary Table S2 in the main text. During these changes, we changed from Supplementary Table S2 to Table 3.
Also, we compared the initial treatment of patients in MPA patients who have progressed to ESRD and those without ESRD. Although no differences were observed in the doses of prednisolone, or the doses of total intravenous cyclophosphamide, or the frequency of administration of rituximab and immunosuppressant therapy between the two groups, the frequency of administration of methylprednisolone pulse therapy was higher in MPA patients who have progressed to ESRD compared to those without ESRD (P = 0.013). We added those sentences in the result sections. (Page 4 Line 116-120)
Reviewer 2 Report
Matsuda et al submit an original research article entitled "Serum complement C4 levels are a useful biomarker for predicting end-stage renal disease in microscopic polyangiitis". In the context of end-stage renal disease (ESRD) in microscopic polyangiitis (MPA), they use a cohort of 74 patients to measure Serum C4 levels and show that levels were significantly higher in MPA patients with ESRD than in those without ESRD. Also Serum C4 levels were significantly positively correlated with serum creatinine levels and kidney Birmingham vasculitis activity score.
Question: there is a huge difference between MPA with ESRD (N=12) and MPA without ESRD(64). Is this representative of the general MPA patient population?
The 5-year ESRD-free rate was 96.2% in the patients 145 without risk factors, 60.5% in the patients with one risk factor, and 0% in those with two 146 risk factors. Does this reflect previously published studies? This should be discussed.
The results found using C4 levels should be discussed and comparred to other new biomarkers described in litterature (N-Gal, microRNAs...)
Author Response
Each response is written following the reviewers’ comments and all revisions are indicated in red font in the revised manuscript. Numbers of page and line correspond to track version.
Comments of Reviewer #2
- There is a huge difference between MPA with ESRD (N=12) and MPA without ESRD (64). Is this representative of the general MPA patient population?
Response: Thank you very much for your important comments. In this study, 16.2% of MPA developed ESRD during follow-up, and cumulative renal survival rate was 77% at 5 years. Furuta, et al previously reported that cumulative renal survival rates in MPA were 76.6%, 78.3%, and 73.6% in UK, Japan, and the EUVAS database, respectively. (Furuta S, et al. J Rheumatol. 2014;41(2):325-33.) Also, in the Irish Rare Kidney Disease Registry, renal survival in MPA-dominant AAV was 79% at 5 years. (Scott J, et al. Sci Rep. 2021;11(1):13080.) Furthermore, registries in Sweden showed that 5 year- survival rate was 70.2% in patients with MPA. (Mohammad AJ. J Rheumatol. 2014;41:1366-73.) These data are consistent with our results, and we thought our data is representative of the general MPA patient population. We added them on the result section and discussion section. (Page 4 Line 101, Page 10 Line 233-236)
- The 5-year ESRD-free rate was 96.2% in the patients without risk factors, 60.5% in the patients with one risk factor, and 0% in those with two risk factors. Does this reflect previously published studies? This should be discussed.
Response: Thank you very much for your helpful comments. In our study, we divided MPA patients into three subgroups based on the number of risk factors, including serum C4 levels and serum creatinine levels, and we found that the ESRD rate increased as the number of risk factors increased. Previous reports support our results, showing that serum creatinine level is a useful marker for predicting ESRD in MPA. (Shi J. BMC Nephrol. 2019;20(1):339. Chen Y, et al. Clin J Am Soc Nephrol. 2017;12(3):417-425.) On the contrary, it has not been elucidated that serum C4 levels can predict the development of ESRD in patients with MPA, but several reports showed that stratification of patients using serum C4 levels was useful for predicting ESRD in other renal diseases. First, Liu, et al. reported that patients with idiopathic membranous nephropathy were divided into three groups according to the serum C4 levels, and they found that the cumulative incidence of renal survival was significantly lower in patients with high C4 levels compared to patients with low and median C4 levels. (Liu J, et al. Front Immunol. 2022;13:896654) Second, Duan, et al. reported that patients with diabetic nephropathy with higher levels of serum C4 had significantly faster deterioration of renal function compared to those with lower levels of serum C4. (Duan S, Front Immunol. 2020;11:2073) Third, Bi, et al. showed that patients with IgA nephropathy with high serum C4 levels had poorer prognosis compared to those with low and normal serum C4 levels. (Bi TD, et al. BMC Nephrol. 2019;20:244.) These results may support our findings, and classification of MPA patients using serum Cr levels and C4 levels at baseline may be useful for predicting ESRD in MPA patients. However, further studies are needed to elucidate the pathogenesis of the association of high C4 levels with the risk of developing renal injury in MPA-GN. We added them in discussion section. (Page 11 Line 278-293)
- The results found using C4 levels should be discussed and comparred to other new biomarkers described in litterature (N-Gal, microRNAs...)
Response: Thank you very much for your helpful suggestions. As you pointed out, there are several biomarkers, including serum N-Gal, microRNAs, and urinary MCP-1, to correlate with renal disease activity in patients with AAV. (Mitsnefes MM, et al. Pediatr Nephrol. 2007;22(1):101-8. Scullion KM, et al. iScience. 2020;24(1):101937. Lieberthal JG, et al. J Rheumatol. 2013;40(5):674-83.) However, there were few reports to show the relationship between biomarkers and ESRD in AAV. (Tam FW, et al. 2004;19(11):2761-8. Park PG, et al. BMC Nephrol. 2022;23(1):288.) Our study showed that the optimal cut-off of baseline serum C4 levels for ESRD progression was set at 29.6 mg/dl, and measuring whether baseline serum C4 level is over this cutoff level may be an invasive and useful method for predicting ESRD in MPA. However, we have not clarified whether serum C4 level is a more sensitive predictor of ESRD compared to other biomarkers in this study. Therefore, further studies are need to prove this. We added them on the discussion section. (Page 11 Line 296-304)
Round 2
Reviewer 1 Report
The authors responded appropriately to my remarks and modified the manuscript accordingly